# A Causal DAG Prior for Synthetic Time-Series Classification Datasets

**Franco Martino O'Rourke** [1]    **Ana Trisovic** [1]    **Dimitris Bertsimas** [1]

## Abstract

A Prior-data fitted Network learns the posterior predictive induced by its training prior; bringing this paradigm to multivariate time-series classification therefore calls for a synthetic generator that produces *complete labelled datasets with temporal structure*. We introduce a causal prior that synthesizes each dataset from a randomly sampled DAG over typed nodes *across two modalities* (tabular attributes and time series), natively producing multivariate, multi-class TSC datasets with cross-modal causal structure across channels, timesteps and labels, a regime not addressed by existing synthetic priors. To validate the prior, we finetune TabPFN v2.5 with minimal adaptations and evaluate on 75 UCR/UEA datasets within TabPFN's operating regime. Finetuning on our generator significantly outperforms both the unmodified upstream model and a tabular-only ablation of the same prior (Wilcoxon signed-rank $p = 3.0 \times 10^{-8}$ on ROC–AUC), isolating the contribution of the cross-modal temporal structure.

## 1. Introduction

**The two-stage status quo in TSC.**   Current foundation-model pipelines for time-series classification (Goswami et al., 2024; Feofanov et al., 2025) follow a *two-stage* recipe: a large transformer is pretrained on unlabelled time-series corpora as an encoder of single series, and a dataset-specific classifier (typically a linear head, a $k$-NN, or TabPFN on the embeddings) is then fit on top of the frozen representations of the training split. Because the encoder ingests one sample at a time, it cannot use the other training samples or their labels as context when choosing which representation to extract, and the two stages optimise different objectives.

[1]Massachusetts Institute of Technology, Cambridge, MA, USA. Correspondence to: Franco Martino O'Rourke <franco03@mit.edu>.

*Proceedings of the $2^{nd}$ ICML Workshop on Foundation Models for Structured Data*, Seoul, South Korea. 2026. Copyright 2026 by the author(s).

**In-context learning as an alternative.**   For tabular data, TabPFN (Hollmann et al., 2023; 2025; Grinsztajn et al., 2025) demonstrates a single-stage alternative: one transformer is pretrained on synthetic datasets sampled from a structural causal prior, and at inference time predicts labels for a query set by conditioning on the entire labelled training set in a single forward pass. The same paradigm, applied to TSC, would ingest $(X, y_{\text{train}}) \in \mathbb{R}^{n \times m \times T} \times \{1, \ldots, K\}^n$ as context and integrate temporal structure, cross-channel interactions and label information from the start. The engine enabling this is a *Bayesian prior-fitted network* (Müller et al., 2022): the model learns the posterior predictive induced by a chosen data generator, so the synthetic prior directly determines the inductive bias of the resulting classifier.

**What is missing: a prior over complete labelled datasets.** Real labelled TSC corpora could in principle train such a model, but publicly available benchmarks are small and narrow in domain; and existing synthetic priors for time series do not generate the object we need. KernelSynth (Ansari et al., 2024) and CauKer (Xie et al., 2025) both sample time series from random Gaussian-process kernels (composed via a structural causal model in CauKer), but neither emits a class label, so neither produces a complete labelled classification dataset. Concurrently with our work, TiCT (Yeh et al., 2025) pre-trains an in-context classifier on synthetic data, but its prior is a binary mixup between two univariate KernelSynth templates: not based on a causal DAG, and not natively multivariate, multi-class or cross-modal at the prior level. What is absent is a prior that samples entire datasets $(X, y)$ by modelling the causal relationships *between two modalities*, tabular attributes and time series. In real systems it is common to find simultaneously: tabular attributes that influence temporal dynamics (*e.g.* soil type $\rightarrow$ humidity$_t$); time series that influence other time series across channels (humidity$_t \rightarrow$ growth$_t$); variables with temporal self-causation (humidity$_t \rightarrow$ humidity$_{t+1}$, humidity$_t \rightarrow$ growth$_{t+1}$); and entire trajectories that determine a tabular outcome (growth$_{1:T} \rightarrow$ yield). Our prior samples a random causal graph over typed nodes so that all of these relationships coexist, exposing the model to the full space of cross-modal structures at training time.

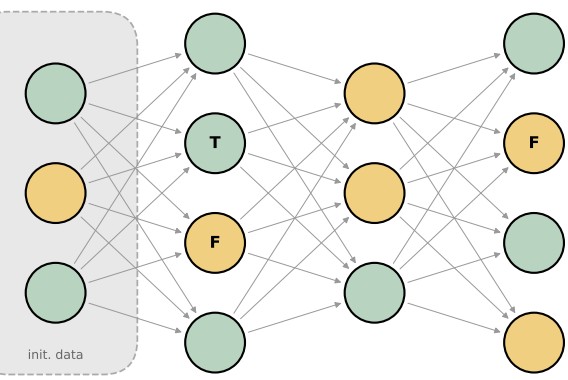

*Figure 1.* Each synthetic dataset is generated from a typed directed acyclic graph (DAG), where **tabular** nodes are scalars ($\in \mathbb{R}$) and **series-valued** nodes are trajectories ($\in \mathbb{R}^T$). Root noise is drawn per sample from random scalar distributions (tabular) or from a Gaussian-process kernel composition fixed per dataset (series-valued). Edges represent computational mechanisms including small neural networks, (causal) Conv1D, and discretisation. Selected series-valued nodes (**F**) define the input $X$; a discretised tabular node (**T**) defines the class label $y$. Optional post-processing is then applied.

**Contribution.** Building on top of the tabular causal prior of Hollmann et al. (2025), we introduce a synthetic causal prior for multivariate TSC datasets that captures tabular, temporal, cross-channel and label structure inside a single directed acyclic graph (DAG).

To probe whether the prior contributes a useful learning signal we finetune TabPFN v2.5, applying only the minimal input/output edits needed to make its tabular pipeline equivalent on time-series data without destroying temporal structure (the goal is not to improve the architecture but to isolate the contribution of the synthetic prior).

## 2. Methodology

### 2.1. Synthetic TSC data based on causal models

We build on top of the causal-DAG prior of Hollmann et al. (2025), extending it so that every sampled dataset is a complete labelled multivariate TSC dataset. Like the tabular prior, our generator is based on structural causal models (SCMs), which here make it possible to model causal relationships *between* two modalities (tabular attributes and time series) inside a single graph. Rather than unrolling a *dynamic* SCM recurrently over time (Boeken & Mooij, 2024), we favour a clean, scalable design: an acyclic graph that injects temporal structure non-recurrently, sidestepping the instability and scaling difficulties that recurrent dynamics can exhibit under an untrained prior (Pascanu et al., 2013).

Our pipeline first samples dataset-level hyperparameters (e.g. dataset size, number of feature channels, series length, difficulty), and then a DAG whose nodes are typed as either *tabular* (a scalar per sample) or *series-valued* (a length-$T$ trajectory per sample). Edges can therefore carry causal effects both within and across modalities (tabular $\rightarrow$ series, series $\rightarrow$ tabular).

To generate each sample, we propagate fresh initialisation data through the DAG's root nodes. Tabular roots are drawn from a random normal or uniform distribution, as in Hollmann et al. (2025). For each series root, a random kernel is sampled *once per dataset* (a composition of Linear/RBF/Periodic base kernels, in the spirit of Kernel-Synth (Ansari et al., 2024)), and per-sample trajectories are then drawn from the zero-mean Gaussian process it defines; see Sec. A.3.

As these data traverse the edges of the computational graph we apply a diverse set of computational mappings, chosen according to the type of the child node. At tabular child nodes the mapping is either a small neural network with a pointwise nonlinearity or a discretisation that turns a numerical output into a categorical one; at series child nodes the mapping is a single random 1-D convolution with nonlinear activation. As an illustration, suppose a series node has four causal parents, two tabular and two series-valued: each tabular parent is replicated along the time axis to yield a constant length-$T$ channel, the two series parents contribute one channel each, and the resulting four-channel input is mapped to a length-$T$ output by a 1-D convolution with causal padding, a randomly sampled dilation and a nonlinear activation, encoding cross-channel mixing and temporal dependence in a single operation. As a second illustration, suppose a tabular node has a single series-valued parent: the parent trajectory is flattened to length $T$ and the tabular node output is a small neural network applied to this flat input, so that a scalar outcome is determined by the entire trajectory of its parent. At each edge we optionally add Gaussian noise; see Sec. A.2 for details.

After traversing the causal graph, we read off a multivariate trajectory $\mathbf{x} \in \mathbb{R}^{m \times T}$ at the sampled series-valued feature nodes and a categorical target $y \in \{1, \dots, K\}$ at a tabular node whose mechanism is a discretisation. Propagating fresh initialisation data through the same graph $n$ times yields a complete labelled TSC dataset $(X, \mathbf{y}) \in \mathbb{R}^{n \times m \times T} \times \{1, \dots, K\}^n$; classification is therefore a read-out on top of the SCM, not a separate generator.

To further expose the model to common data challenges, we apply predictive truncation (variable-length observations), value-level missingness, random temporal granularity (strided pooling and step-repeat) and per-channel value warps drawn from {identity, log, exp, squash, KDI, Kumaraswamy} (Hollmann et al., 2025); see Sec. A.4.

## 2.2. Using TabPFN as a probe of the prior

To probe the learning signal of the prior we finetune the pretrained TabPFN v2.5 backbone (Grinsztajn et al., 2025). We do not modify the backbone; we apply only the minimal input/output edits needed to make TabPFN's tabular pipeline equivalent on inputs of shape $n \times m \times T$ without destroying temporal structure. The aim of these edits is not to improve the architecture, but to keep it on equal footing so that any change in performance reflects the synthetic prior alone.

If a TSC dataset is flattened as if it were tabular it becomes a matrix $X \in \mathbb{R}^{n \times (m \cdot T)}$, with one column per (channel, timestep) pair. TabPFN v2.5 forms input tokens by grouping every $G = 3$ consecutive columns (together with their missingness indicators) and projecting the group through a linear tokeniser. Our first adaptation is that the grouping is forced to respect the channel boundary: we pad the time axis of each channel to the next multiple of $G$ before flattening, so that every token covers $G$ consecutive timesteps of the *same* channel. The flattened input thus becomes a sequence of short temporal *patches* of each channel, in the spirit of PatchTST (Nie et al., 2023).

TabPFN does not use a positional encoding; instead, the tokeniser appends a fixed pseudo-random embedding to every column so the attention layers can tell columns apart. We leave this mechanism unchanged: after finetuning, when the transformer attends to the entire labelled context of $n$ samples, we expect it to be able to discover the correlation structure between patches from the context itself even though the pseudo-random embeddings do not encode column order explicitly. A second adaptation is normalisation: TabPFN's default `normalize_x` step $z$-scores each of the $G$ within-token positions independently across samples, which would wipe out systematic temporal patterns inside a patch; we disable it and instead apply $z$-scoring per channel before the tokeniser, so that continuity within each patch is preserved. The remaining preprocessing is likewise computed per channel rather than per column. Tokeniser weights, transformer blocks and decoder are all inherited verbatim from TabPFN v2.5; we do not modify the ICL objective. At inference we ensemble $e \in \{1, 8\}$ forward passes per dataset, with random channel and class-index permutations between members; full schedule and ensembling details are deferred to Appendices B–C.

## 3. Experiments

### 3.1. Datasets

We evaluate on the subset of UCR (Dau et al., 2019) and UEA (Bagnall et al., 2018) that fits within the operating regime of TabPFN. TabPFN's native classification head is limited to $K \leq 10$ classes (Hollmann et al., 2025), and v2.5 applies per-estimator feature subsampling beyond 500

columns (Grinsztajn et al., 2025); when a multivariate series is flattened to a tabular row, the latter limit applies to the product $m \cdot T$. We therefore retain all UCR and UEA datasets with $K \leq 10$ and $m \cdot T \leq 500$, obtaining 75 datasets. We use the default train/test split supplied with each dataset and report mean per-dataset accuracy and ROC–AUC; for multiclass datasets we report macro-averaged one-vs-rest ROC–AUC. Significance between pairs of models is assessed with a Wilcoxon signed-rank test over the 75 datasets; global ranks are summarised with a critical-difference (CD) diagram (Demšar, 2006; Middlehurst et al., 2024).

### 3.2. Configurations

We compare four configurations, all built around the same TabPFN v2.5 backbone, at two ensemble sizes $e \in \{1, 8\}$:

**(A) Standard TabPFN.** The pretrained model is applied to the series flattened as a tabular row $X \in \mathbb{R}^{n \times (m \cdot T)}$, using the default TabPFN preprocessing and inference pipeline (column-wise `normalize_x`, default tokeniser grouping, default SVD ensembling). No modifications, no finetuning.

**(B) + Our preprocessing.** Same weights as (A), but with the adaptations of Sec. 2.2 (patch-aligned tokenisation, per-channel $z$-scoring, `normalize_x` disabled). Still no finetuning.

**(C) + Finetuning (ours).** (B) further finetuned on datasets drawn from our full generator (Sec. 2.1).

**(D) + Finetuning, tabular-only ablation.** Same setup as (C), but every series-specific component of the generator is disabled: series-valued nodes, GP roots, Conv1D mechanisms, and the time-series augmentations of Sec. A.4, reducing the prior to a purely tabular generator with $T = 1$.

### 3.3. Main result

Table 1 reports mean accuracy and AUC across the 75 benchmarks for the two ensemble sizes most commonly used in the TabPFN literature. Configuration (C) achieves the best accuracy and AUC at both $e = 1$ and $e = 8$. Our custom inference alone (B) *loses* a small amount of accuracy relative to (A), which is expected: our inference preset trades away an SVD column-whitening step that helps standard tabular data. Finetuning on the full generator (C) more than recovers the gap and overtakes the upstream model; finetuning on the tabular-only ablation (D) does *not*.

Figure 2 visualises the same picture as a critical-difference ranking over all eight configurations (four models × two ensemble sizes): both variants of (C) sit clearly ahead of (A), (B) and (D).

*Table 1.* Mean accuracy and ROC–AUC on the 75 datasets. Best per column in **bold**. (C) uses our full prior; (D) is the tabular-only ablation defined in Sec. 3.2.

| | $e{=}1$ | | $e{=}8$ | |
|---|---|---|---|---|
| Model | Acc. | AUC | Acc. | AUC |
| (A) Standard TabPFN | .8436 | .9372 | .8525 | .9405 |
| (B) Pretrained + our inf. | .8378 | .9368 | .8489 | .9399 |
| (D) Finetuned, tab. gen. | .8366 | .9354 | .8493 | .9393 |
| (C) Finetuned, *ours* | **.8559** | **.9440** | **.8575** | **.9444** |

*Table 2.* Wilcoxon signed-rank tests across the 75 datasets, on both metrics. "Wins" (computed on ROC–AUC) is the share of datasets on which model A strictly beats model B.

| A vs B | Wins A | Wins B | $p_{\text{AUC}}$ | $p_{\text{Acc}}$ |
|---|---|---|---|---|
| $(C_{e=1})$ vs $(D_{e=1})$ | 73.3% | 12.0% | $3.0{\times}10^{-8}$ | $1.1{\times}10^{-6}$ |
| $(C_{e=8})$ vs $(D_{e=8})$ | 66.7% | 14.7% | $1.8{\times}10^{-7}$ | $8.3{\times}10^{-4}$ |
| $(C_{e=1})$ vs $(A_{e=1})$ | 64.0% | 22.7% | $4.4{\times}10^{-4}$ | $4.1{\times}10^{-3}$ |
| $(C_{e=1})$ vs $(B_{e=1})$ | 70.7% | 14.7% | $1.2{\times}10^{-6}$ | $1.5{\times}10^{-6}$ |
| $(D_{e=1})$ vs $(A_{e=1})$ | 38.7% | 48.0% | 0.10 | 0.20 |
| $(D_{e=1})$ vs $(B_{e=1})$ | 24.0% | 57.3% | $1.7{\times}10^{-4}$ | 0.43 |

**Critical difference — AUC (N=75, Wilcoxon+Holm α=0.05)**

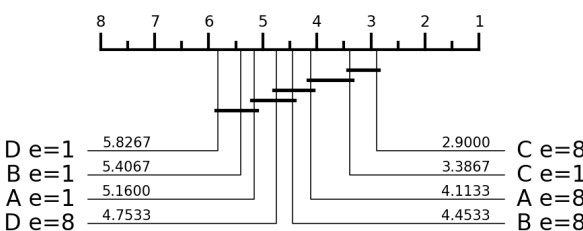

*Figure 2.* Critical-difference diagram (ROC–AUC) across all eight evaluated configurations, 75 datasets. Lower rank is better. The two variants of (C) (ours) sit clearly ahead of (A), (B) and (D).

### 3.4. Ablation: are the series nodes necessary?

The central question is whether the *series* mechanisms of the generator (the GP-KernelSynth roots and the Conv1D interior nodes) actually contribute signal, or whether finetuning on any DAG prior that matches the input shape would suffice. Table 2 reports Wilcoxon signed-rank tests on both mean ROC–AUC and mean accuracy across the 75 datasets.

The results split into two consistent patterns across both metrics. *First*, the full-generator model (C) significantly outperforms every other configuration: it beats the tabular ablation (D) at $p_{\text{AUC}}{=}3.0{\times}10^{-8}$, $p_{\text{Acc}}{=}1.1{\times}10^{-6}$, and the effect is preserved at $e{=}8$ ($p_{\text{AUC}}{=}1.8{\times}10^{-7}$, $p_{\text{Acc}}{=}8.3{\times}10^{-4}$), so the advantage is stable across ensemble sizes. *Second*, the tabular ablation (D) gives no positive signal over the baselines: it is statistically indistinguishable from the upstream model (A) on both metrics ($p_{\text{AUC}}{=}0.10$, $p_{\text{Acc}}{=}0.20$), and on AUC is even outperformed by the same backbone *without* finetuning (B) ($p_{\text{AUC}}{=}1.7{\times}10^{-4}$). Finetuning on a tabular-only DAG prior thus does not help, and can even hurt relative to simply applying our patch-aligned inference. The gain therefore cannot be attributed to the finetuning recipe or the input/output adaptations alone; it is the series mechanisms of the prior that supply the signal.

## 4. Conclusion

We presented a causal prior for synthetic labelled TSC datasets that types nodes in two modalities (tabular and series-valued) and models causal effects *between* them inside a single structural causal model. Using TabPFN v2.5 as a probe, with only the minimal input/output edits required to preserve temporal structure, we showed that finetuning on this prior improves over both the unmodified upstream model and a tabular-only ablation of the same generator at the $3{\times}10^{-8}$ level on ROC–AUC, isolating the contribution of the cross-modal temporal structure.

**Extension to richer structured data.** Because the prior operates at the level of *node types*, the construction is not specific to time series. A `spatial` node type whose mechanism is a 2-D convolution would yield image datasets; a `spatio-temporal` type using a 3-D convolution would yield video-like tensors; and mixing node types along the same DAG would yield truly multimodal datasets, where the input contains both tabular attributes and time series and the label depends on a combination of both. To our knowledge, no existing synthetic prior models causal relationships *between* modalities of this kind.

**Limitations.** This is a paper about the *generator*: we did not redesign the backbone, so the resulting probe is not a state-of-the-art TSC foundation model and we do not benchmark against full leaderboards; different patch sizes, a different tokeniser, or pretraining from scratch on this prior are all likely to produce a stronger model. Evaluation is further restricted to the $(m \cdot T \le 500,\ K \le 10)$ operating regime of TabPFN, which excludes long or high-dimensional series, and the results only attest to the incremental value of the series mechanisms over a tabular prior, not to absolute TSC performance. Finally, while the overall improvement supports the effectiveness of causal DAGs that mix tabular and series-valued nodes, performance degrades on some individual datasets, warranting further attention to keep improving the prior coverage.

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

# A. Details on the causal generative process

This section gives the technical details of the generator of Sec. 2. An SCM $\mathcal{G} := (\mathbf{Z}, \varepsilon)$ is a collection $\mathbf{Z} := (z_1, \ldots, z_M)$ of structural assignments $z_i = f_i(z_{\text{PA}_\mathcal{G}(i)}, \varepsilon_i)$, where $\text{PA}_\mathcal{G}(i)$ are the direct causes of $z_i$ in a DAG $\mathcal{G}$, $f_i$ is a deterministic mechanism, and $\varepsilon_i$ is exogenous noise. Every component below is defined by a distribution; we use U, LogU and $\text{LogU}_\mathbb{Z}$ for uniform, log-uniform and log-uniform integer distributions.

## A.1. Graph structure sampling

The structural causal model underlying each dataset is based on a DAG $\mathcal{G}$. We sample $\mathcal{G}$ as a layered feed-forward graph: a root layer of width $d_0 \sim \text{LogU}_\mathbb{Z}[a_d, b_d]$ is followed by $L \sim \text{LogU}_\mathbb{Z}[a_L, b_L]$ hidden layers of widths $d_\ell \sim \text{LogU}_\mathbb{Z}[a_w, b_w]$. Every node in a layer is initially connected to every node in the previous layer, and edges are then dropped independently with probability $p_{\text{drop}} \sim \text{U}[\alpha_{\text{drop}}, \beta_{\text{drop}}]$. Disjoint subgraphs are allowed and lead to features that are marginally independent of the target if they are not connected to the target node, reflecting real-world scenarios with uninformative predictors.

The node type $\tau_i \in \{\texttt{tabular}, \texttt{series}\}$ is sampled independently per node: with probability $\pi_\text{S}^\text{root}$ at the root layer and $\pi_\text{S}^\text{hidden} \sim \text{U}[\alpha_\text{S}, \beta_\text{S}]$ at hidden layers the node is a $\texttt{series}$ node; otherwise it is $\texttt{tabular}$. Three structural guarantees are enforced by resampling: (i) $\mathcal{G}$ contains at least one series root, at least one hidden series node and at least one tabular node whose mechanism is a discretisation, so a classification target exists; (ii) every non-root node has at least one parent; (iii) every series node has at least one series parent, so series ancestry cannot be broken by a tabular bottleneck.

## A.2. Computational edge mappings

In our implementation, each tabular SCM node and sample is represented as a scalar in $\mathbb{R}$, and each series SCM node and sample as a vector in $\mathbb{R}^T$. When propagating data through the SCM, the deterministic function $f_i$ at each non-root node maps the concatenation of the parent outputs to its own output using one of three types of computational modules, chosen according to the node type:

*1. Small neural networks (tabular nodes).* We initialise a weight vector $\mathbf{w} \in \mathbb{R}^{d_i}$ using Xavier initialisation and apply the linear transformation $\mathbf{w}^\top \mathbf{x} + b$ to the input vector $\mathbf{x}$, where $b \in \mathbb{R}$ is a bias. After the linear projection, we apply an element-wise nonlinear activation function $\sigma : \mathbb{R} \to \mathbb{R}$, randomly sampled from a bank including identity, logarithm, sigmoid, absolute value, sine, hyperbolic tangent, squaring, power functions, smooth ReLU, step function and modulo operation. Tabular parents contribute a single scalar to $\mathbf{x}$; series parents are flattened along time and contribute $T$ entries each.

*2. Categorical feature discretisation (tabular nodes).* To generate categorical features from the numerical vectors at a node, we map the input vector to the index of the nearest neighbour in a set of $K$ randomly sampled prototype vectors $\{\mathbf{p}_1, \ldots, \mathbf{p}_K\}$ for a feature with $K$ categories, $\kappa_i = \arg\min_c \|\mathbf{x} - \mathbf{p}_c\|$. The discrete index $\kappa_i$ will be observed in the feature set as a categorical feature, or read off as the label if this node is selected as the target (Sec. A.5). To further use these discrete class assignments in the computational graph they are re-embedded as continuous scalars: we sample a per-class embedding $v_c \sim \mathcal{N}(0, 1)$ once per node and propagate $z_i = v_{\kappa_i}$ to descendants. The number of categories $K$ is drawn from $\text{LogU}_\mathbb{Z}[a_K, b_K]$.

*3. One-dimensional convolutions (series nodes).* To encode cross-channel and temporal dependencies, series nodes apply a single Conv1D. Let $\mathbf{x} \in \mathbb{R}^{c_\text{in} \times T}$ be the concatenation of the parent channels along the channel axis: scalar tabular parents are first broadcast to a constant length-$T$ channel, and series parents contribute one channel each. The series node output is

$$z_i(t) = \sigma\left(\sum_{c=1}^{c_\text{in}} \sum_{\ell=1}^{K} W_{c,\ell}\, \mathbf{x}_c(t - D(\ell - 1)) + b\right), \qquad (1)$$

where the kernel length $K$ is drawn uniformly from a small set of odd lengths plus the unit kernel, the dilation $D = \lfloor 2^u \rfloor$ with $u \sim \text{U}[0, \log_2 \frac{T-1}{K-1}]$, the weights $W_{c,\ell} \sim \mathcal{N}(0, 1)$ are mean-centred, and the bias $b \sim \text{U}[-\gamma, \gamma]$. A dataset-level padding mode (causal or centred) is drawn once per dataset and shared by all series nodes, giving the generator a consistent notion of temporal directionality. The activation $\sigma$ is drawn from the same bank as for tabular nodes, with the identity up-weighted so that a recognisable temporal signal is preserved through deeper graphs.

*4. Noise injection.* At each non-root node we optionally add Gaussian noise drawn from $\mathcal{N}(0, \sigma_\varepsilon^2 \mathbf{I})$ with probability $p_\varepsilon$.

### A.3. Initialization data sampling

For each to-be-generated sample, we randomly generate initialisation data $\varepsilon$ that is inserted at the DAG root nodes and then propagated through the computational graph. The noise is sampled according to the type of each root node:

*1. Tabular roots.* $\varepsilon \sim \mathcal{N}(0, \sigma_\varepsilon^2)$ or $\varepsilon \sim \mathrm{U}[-a, a]$, one family drawn per root. If a tabular root is additionally assigned the discretisation mechanism, its scalar is first mapped to a categorical index and re-embedded as a scalar before being propagated.

*2. Series roots.* The kernel $k$ is sampled *once per dataset* as a random composition of $J \sim \mathrm{LogU}_{\mathbb{Z}}[a_J, b_J]$ base kernels drawn uniformly from a bank $\{k_{\mathrm{Lin}}, k_{\mathrm{RBF}}, k_{\mathrm{Per}}\}$; the composition is built by iteratively picking two kernels from the current pool and combining them with either a sum or a product, until a single kernel remains. For each sample, the entire length-$T$ trajectory is then drawn from a Gaussian process,

$$\mathbf{f} \sim \mathcal{GP}(0, k(t, t')), \qquad t \in \left\{ \tfrac{0}{T-1}, \dots, \tfrac{T-1}{T-1} \right\}. \tag{2}$$

The kernel bank and compositional operators follow KernelSynth (Ansari et al., 2024).

### A.4. Post-processing / augmentation

Each dataset is post-processed randomly with one or more of the following operations:

*1. Per-channel value warps.* For some datasets, we apply a per-channel transform drawn independently from {identity, log, exp, squash, KDI, Kumaraswamy}, introducing nonlinear distortions and scale differences. The Kumaraswamy warp uses the Kumaraswamy CDF as in Hollmann et al. (2025).

*2. Temporal granularity.* A single transform is drawn per dataset from {identity, strided pooling, step-repeat}. Strided pooling lowers the sampling rate by averaging over consecutive timesteps; step-repeat artificially lowers the rate by emitting each value several times. This exposes the model to datasets measured at different sampling rates of the same underlying process.

*3. Predictive truncation.* We designate a fraction of samples as truncated: their observed series is cut to a random sub-horizon $t_{\mathrm{obs}} < T$ while the target is still read from the full trajectory of length $T$. This trains the model to produce predictions from incomplete observations.

*4. Value-level missing completely at random.* To introduce scenarios for dynamic imputation, a fraction $\rho_{\mathrm{miss}}$ of observations in each series is masked as missing, independently of the data values.

Every sampled dataset is constrained to the PFN-eligibility region $K \leq K_{\max}, T \leq T_{\max}, m \leq m_{\max}, m \cdot T \leq (mT)_{\max}$, $n_{\mathrm{train}} \leq n_{\max}$, matching the regime of our probe.

### A.5. Target generation

To generate the classification target, we select uniformly at random one tabular node whose mechanism is a discretisation and set $y_s = \kappa_{\star, s}$. The number of classes is therefore bounded by the number of prototypes at that node; we cap it at 10 classes.

## B. Training details

We finetune a pretrained TabPFN v2.5 backbone (Grinsztajn et al., 2025) on datasets drawn from the generator of App. A. The architecture is kept essentially unchanged; three adaptations are made at the input/output boundary so that an $n \times m \times T$ dataset can be ingested. First, after sampling a dataset we reshape it as $X \in \mathbb{R}^{n \times (m \cdot T)}$ and pad the time axis to the next multiple of the model's column-group size $G$. TabPFN forms a token by grouping $G$ consecutive columns; padding in this way guarantees that each token covers $G$ consecutive timesteps of the *same* channel and that channels never mix inside a token. To preserve this temporal structure during inference-time ensembling, we also replace the default column shuffling used by TabPFN with *channel shuffling*, i.e., we permute entire channels instead of individual columns so that consecutive timesteps always remain adjacent. Second, we apply per-channel $z$-scoring before the transformer, and we disable the encoder's internal `normalize_x` step, which would $z$-score each of the $G$ within-token positions across samples and destroy systematic temporal patterns. Tokeniser, transformer blocks and decoder are inherited verbatim.

The training loss is the cross-entropy between the targets of held-out samples of a synthetic dataset and the model prediction: for a test set $(X_{\text{test}}, y_{\text{test}}) = \mathcal{D}_{\text{test}}$ drawn from our prior and the associated training context $\mathcal{D}_{\text{train}}$, $\mathcal{L} = \mathbb{E}[-\log q_\theta(y_{\text{test}} \mid X_{\text{test}}, \mathcal{D}_{\text{train}})]$ (Müller et al., 2022). We use AdamW with linear warmup followed by cosine annealing.

## C. Inference details

At inference we feed a full labelled training set $(X_{\text{train}}, y_{\text{train}}) \in \mathbb{R}^{n \times m \times T} \times \{1, \ldots, K\}^n$ together with a query batch $X_{\text{test}} \in \mathbb{R}^{n_{\text{test}} \times m \times T}$ as one context, and the model returns a posterior over $y_{\text{test}}$ in a single forward pass. As our model is not fully permutation invariant, for each ensemble member we shuffle the feature-channel order and, for classification, we additionally permute the class labels, approximating order invariance. We ensemble $e \in \{1, 8\}$ forward passes per dataset, averaging predictions across members. We do not apply TabPFN's SVD column-whitening preset at inference, which is tuned for tabular columns rather than for the temporal columns of a single channel. Instead, we concatenate a pooled representation of each channel, providing global channel-level context while preserving the original temporal structure of the series.

## D. Generation cost

On a single CPU core, generating 100 datasets takes approximately 28 s (mean 0.24 s per dataset, median 0.03 s, max 8.3 s). The high variance stems from the log-uniform distributions over $n \in [30, 1400]$ and $T \in [6, 2000]$: cost scales with $n \cdot T$, so outliers arise when both are drawn large.

## E. Per-dataset results

*Table 3.* Per-dataset ROC–AUC of the eight evaluated configurations on the 75 PFN-eligible UCR/UEA datasets. Winner per dataset in **bold**.

| Dataset | A $e{=}1$ | A $e{=}8$ | B $e{=}1$ | B $e{=}8$ | C $e{=}1$ | C $e{=}8$ | D $e{=}1$ | D $e{=}8$ |
|---|---|---|---|---|---|---|---|---|
| AllGestureWiimoteX | .896 | .925 | .874 | .922 | .930 | **.937** | .864 | .920 |
| AllGestureWiimoteY | .912 | .940 | .951 | .952 | .954 | **.954** | .950 | .952 |
| AllGestureWiimoteZ | .913 | **.919** | .888 | .902 | .918 | .918 | .883 | .901 |
| ArrowHead | .909 | .910 | .935 | .927 | **.944** | .936 | .932 | .924 |
| BME | **1.000** | **1.000** | **1.000** | **1.000** | **1.000** | **1.000** | **1.000** | **1.000** |
| Beef | .981 | **.986** | .968 | .979 | .976 | .985 | .968 | .979 |
| CBF | .994 | .995 | .995 | .995 | **.998** | .997 | .995 | .995 |
| Chinatown | .996 | .995 | .996 | .996 | .994 | .996 | .996 | **.996** |
| ChlorineConcentration | .996 | **.999** | .991 | .994 | .996 | .996 | .991 | .994 |
| Coffee | **1.000** | **1.000** | **1.000** | **1.000** | **1.000** | **1.000** | **1.000** | **1.000** |
| DiatomSizeReduction | 1.000 | **1.000** | 1.000 | 1.000 | 1.000 | 1.000 | .999 | .999 |
| DistalPhalanxOutlineAgeGroup | .894 | .902 | .900 | .903 | .904 | **.905** | .900 | .903 |
| DistalPhalanxOutlineCorrect | .855 | .862 | **.884** | .878 | .877 | .878 | .880 | .877 |
| DistalPhalanxTW | .895 | **.909** | .885 | .890 | .890 | .895 | .883 | .889 |
| DodgerLoopDay | .903 | .909 | .821 | .897 | .885 | **.916** | .822 | .898 |
| DodgerLoopGame | .841 | .820 | **.904** | .868 | .891 | .875 | .903 | .873 |
| DodgerLoopWeekend | .986 | .987 | .989 | .990 | .990 | **.991** | .989 | **.991** |
| ECG200 | **.939** | .937 | .923 | .927 | .934 | .934 | .924 | .928 |
| ECG5000 | .948 | .950 | .946 | .947 | **.951** | .951 | .945 | .946 |
| ECGFiveDays | .981 | **.997** | .950 | .960 | .979 | .983 | .939 | .954 |
| ERing | .992 | .995 | .990 | .993 | .995 | **.996** | .990 | .993 |
| ElectricDevices | .891 | .902 | .892 | **.903** | .897 | .898 | .892 | .903 |
| FaceFour | **.992** | .991 | .990 | .990 | .986 | .987 | .989 | .990 |
| Fish | **.995** | .994 | .989 | .994 | .992 | .994 | .976 | .993 |
| FordA | .976 | **.977** | .925 | .943 | .944 | .951 | .924 | .944 |
| FordB | **.874** | .869 | .737 | .732 | .727 | .730 | .734 | .731 |
| FreezerRegularTrain | 1.000 | 1.000 | 1.000 | 1.000 | 1.000 | 1.000 | **1.000** | 1.000 |
| FreezerSmallTrain | .990 | .993 | .999 | .998 | **1.000** | 1.000 | .997 | .997 |
| GesturePebbleZ1 | .981 | .983 | .983 | .986 | **.986** | .985 | .983 | .985 |
| GesturePebbleZ2 | .954 | .954 | .946 | .959 | **.972** | .971 | .946 | .957 |
| GunPoint | .995 | .990 | .982 | .988 | .996 | **.997** | .981 | .989 |
| GunPointAgeSpan | **1.000** | **1.000** | 1.000 | **1.000** | **1.000** | **1.000** | **1.000** | **1.000** |
| GunPointMaleVersusFemale | **1.000** | **1.000** | **1.000** | **1.000** | **1.000** | **1.000** | **1.000** | **1.000** |
| GunPointOldVersusYoung | **1.000** | **1.000** | **1.000** | **1.000** | **1.000** | **1.000** | **1.000** | **1.000** |
| Ham | .834 | .816 | .816 | .814 | **.845** | .840 | .821 | .816 |
| ItalyPowerDemand | .994 | .994 | **.994** | .994 | .994 | .993 | .994 | .993 |
| JapaneseVowels | 1.000 | 1.000 | 1.000 | 1.000 | **1.000** | 1.000 | 1.000 | 1.000 |
| Lightning7 | .935 | .943 | .945 | .946 | .961 | **.962** | .942 | .945 |
| Meat | **1.000** | **1.000** | **1.000** | **1.000** | **1.000** | **1.000** | **1.000** | **1.000** |
| MedicalImages | .975 | .978 | .977 | .982 | .984 | **.984** | .977 | .982 |

*continued on next page*

*(continued)*

| Dataset | A $e$=1 | A $e$=8 | B $e$=1 | B $e$=8 | C $e$=1 | C $e$=8 | D $e$=1 | D $e$=8 |
|---|---|---|---|---|---|---|---|---|
| MelbournePedestrian | .999 | 1.000 | .999 | 1.000 | .999 | **1.000** | .999 | .999 |
| MiddlePhalanxOutlineAgeGroup | **.643** | .640 | .640 | .634 | .642 | .641 | .640 | .630 |
| MiddlePhalanxOutlineCorrect | .927 | .929 | .917 | .915 | .931 | **.932** | .912 | .913 |
| MiddlePhalanxTW | .789 | **.810** | .775 | .809 | .799 | .803 | .768 | .806 |
| MoteStrain | .959 | **.961** | .948 | .950 | .960 | .960 | .949 | .950 |
| OSULeaf | .905 | .911 | .896 | .895 | **.918** | .911 | .894 | .893 |
| PenDigits | 1.000 | 1.000 | 1.000 | 1.000 | **1.000** | 1.000 | 1.000 | 1.000 |
| PhalangesOutlinesCorrect | .919 | **.922** | .907 | .912 | .914 | .919 | .904 | .910 |
| PickupGestureWiimoteZ | .947 | .923 | .946 | .942 | **.953** | .950 | .944 | .944 |
| Plane | **1.000** | **1.000** | **1.000** | **1.000** | **1.000** | **1.000** | **1.000** | **1.000** |
| PowerCons | **1.000** | **1.000** | **1.000** | **1.000** | **1.000** | **1.000** | **1.000** | **1.000** |
| ProximalPhalanxOutlineAgeGroup | .939 | .941 | .936 | .941 | .941 | **.943** | .936 | .938 |
| ProximalPhalanxOutlineCorrect | .953 | **.966** | .960 | .955 | .954 | .961 | .958 | .954 |
| ProximalPhalanxTW | .943 | .936 | **.948** | .946 | .935 | .942 | .948 | .947 |
| RacketSports | .967 | .968 | .971 | .973 | .974 | **.974** | .972 | .973 |
| ShakeGestureWiimoteZ | .971 | .975 | .987 | .994 | .996 | **.998** | .984 | .994 |
| ShapeletSim | .459 | .468 | .466 | **.473** | .470 | .468 | .468 | .466 |
| SmoothSubspace | **1.000** | **1.000** | **1.000** | **1.000** | **1.000** | **1.000** | **1.000** | **1.000** |
| SonyAIBORobotSurface1 | .948 | **.959** | .933 | .939 | .958 | .958 | .935 | .941 |
| SonyAIBORobotSurface2 | .900 | .909 | .902 | .901 | **.912** | .911 | .904 | .903 |
| Strawberry | .995 | .995 | .995 | .994 | **.996** | .995 | .995 | .994 |
| Symbols | **.990** | .982 | .981 | .983 | .987 | .987 | .977 | .981 |
| SyntheticControl | 1.000 | 1.000 | **1.000** | **1.000** | 1.000 | **1.000** | **1.000** | **1.000** |
| ToeSegmentation1 | .602 | .612 | **.733** | .671 | .714 | .653 | .713 | .661 |
| ToeSegmentation2 | .759 | .795 | .945 | .954 | .949 | .953 | .945 | **.954** |
| Trace | .998 | .998 | **1.000** | **1.000** | **1.000** | **1.000** | **1.000** | **1.000** |
| TwoLeadECG | .971 | .988 | .975 | .971 | **.992** | .991 | .979 | .974 |
| TwoPatterns | 1.000 | **1.000** | 1.000 | 1.000 | 1.000 | **1.000** | 1.000 | 1.000 |
| UMD | **1.000** | **1.000** | **1.000** | **1.000** | **1.000** | **1.000** | **1.000** | **1.000** |
| UWaveGestureLibraryX | .968 | .970 | .968 | .970 | .970 | **.972** | .968 | .970 |
| UWaveGestureLibraryY | .951 | .954 | .951 | .955 | .951 | **.955** | .951 | .955 |
| UWaveGestureLibraryZ | .961 | **.963** | .957 | .962 | .961 | .962 | .957 | .963 |
| Wafer | 1.000 | 1.000 | 1.000 | 1.000 | **1.000** | 1.000 | 1.000 | 1.000 |
| Wine | .768 | .801 | .724 | .771 | .796 | **.809** | .719 | .760 |
| Yoga | **.945** | .945 | .931 | .935 | .935 | .944 | .930 | .935 |
| *Mean* | .937 | .941 | .937 | .940 | .944 | **.944** | .935 | .939 |

*Table 4.* Per-dataset accuracy of the eight evaluated configurations on the 75 PFN-eligible UCR/UEA datasets. Winner per dataset in **bold**.

| Dataset | A $e$=1 | A $e$=8 | B $e$=1 | B $e$=8 | C $e$=1 | C $e$=8 | D $e$=1 | D $e$=8 |
|---|---|---|---|---|---|---|---|---|
| AllGestureWiimoteX | .494 | .604 | .479 | .590 | .574 | **.631** | .460 | .597 |
| AllGestureWiimoteY | .583 | .664 | .636 | .694 | .667 | **.709** | .626 | .701 |
| AllGestureWiimoteZ | .554 | **.589** | .459 | .514 | .560 | .553 | .460 | .511 |
| ArrowHead | .703 | .726 | .766 | .743 | **.794** | .777 | .771 | .743 |
| BME | .993 | **1.000** | **1.000** | **1.000** | **1.000** | **1.000** | **1.000** | **1.000** |
| Beef | .867 | **.933** | .800 | .867 | .867 | .867 | .867 | .867 |
| CBF | .932 | .929 | .913 | .916 | .929 | **.937** | .913 | .914 |
| Chinatown | **.988** | .985 | .985 | .983 | .983 | .985 | .985 | .985 |
| ChlorineConcentration | .958 | **.973** | .952 | .959 | .965 | .968 | .953 | .958 |
| Coffee | **1.000** | **1.000** | **1.000** | **1.000** | **1.000** | **1.000** | **1.000** | **1.000** |
| DiatomSizeReduction | **.971** | **.971** | .964 | .967 | **.971** | **.971** | .964 | .967 |
| DistalPhalanxOutlineAgeGroup | .719 | .741 | .727 | .770 | .755 | .734 | .748 | **.784** |
| DistalPhalanxOutlineCorrect | .793 | .786 | .797 | **.801** | **.801** | **.801** | .790 | .797 |
| DistalPhalanxTW | .669 | **.691** | .655 | .647 | .655 | .662 | .640 | .655 |
| DodgerLoopDay | .623 | **.636** | .442 | .584 | .597 | **.636** | .442 | .584 |
| DodgerLoopGame | .748 | .748 | **.811** | .740 | .787 | .780 | .803 | .740 |
| DodgerLoopWeekend | **.984** | **.984** | **.984** | **.984** | **.984** | **.984** | **.984** | **.984** |
| ECG200 | **.880** | **.880** | .870 | .870 | **.880** | .880 | .860 | .870 |
| ECG5000 | .943 | .941 | .941 | .941 | **.944** | .944 | .939 | .938 |
| ECGFiveDays | .875 | **.967** | .863 | .872 | .906 | .923 | .848 | .862 |
| ERing | .896 | .919 | .881 | .911 | **.933** | .930 | .889 | .911 |
| ElectricDevices | .700 | .715 | .717 | **.740** | .727 | .737 | .718 | .739 |
| FaceFour | .909 | **.932** | .920 | **.932** | .920 | **.932** | .920 | **.932** |
| Fish | **.909** | .891 | .874 | .897 | .880 | **.909** | .783 | .903 |
| FordA | .913 | **.922** | .842 | .869 | .856 | .852 | .838 | .863 |
| FordB | .784 | **.786** | .659 | .600 | .604 | .585 | .658 | .600 |
| FreezerRegularTrain | .997 | .999 | **.999** | .999 | .998 | .998 | **.999** | .999 |
| FreezerSmallTrain | .932 | .936 | .949 | .951 | **.979** | .964 | .942 | .944 |
| GesturePebbleZ1 | .831 | .831 | .843 | .837 | **.849** | **.849** | .843 | .837 |
| GesturePebbleZ2 | .747 | .772 | .753 | **.785** | **.785** | .766 | .759 | **.785** |
| GunPoint | .953 | .940 | .913 | .913 | **.967** | **.967** | .913 | .927 |
| GunPointAgeSpan | .987 | **.997** | .994 | .994 | .994 | **.997** | .994 | .994 |
| GunPointMaleVersusFemale | **1.000** | **1.000** | .997 | .997 | **1.000** | **1.000** | .997 | .997 |

*(continued)*

| Dataset | A $e{=}1$ | A $e{=}8$ | B $e{=}1$ | B $e{=}8$ | C $e{=}1$ | C $e{=}8$ | D $e{=}1$ | D $e{=}8$ |
|---|---|---|---|---|---|---|---|---|
| GunPointOldVersusYoung | **1.000** | **1.000** | **1.000** | **1.000** | **1.000** | **1.000** | **1.000** | **1.000** |
| Ham | **.781** | .733 | .752 | .733 | .743 | .762 | .762 | .724 |
| ItalyPowerDemand | .973 | .972 | .971 | .973 | .965 | .968 | .970 | **.975** |
| JapaneseVowels | .986 | **.989** | .981 | .984 | .984 | .984 | .981 | .984 |
| Lightning7 | .699 | .671 | .658 | .740 | .712 | .740 | .644 | **.767** |
| Meat | **1.000** | **1.000** | .983 | .983 | .967 | .983 | .967 | .983 |
| MedicalImages | .803 | .800 | .843 | .838 | .847 | .846 | .849 | **.851** |
| MelbournePedestrian | .979 | .979 | .977 | .980 | .978 | **.981** | .977 | .980 |
| MiddlePhalanxOutlineAgeGroup | .584 | .617 | .552 | .578 | **.643** | .630 | .552 | .571 |
| MiddlePhalanxOutlineCorrect | .821 | **.849** | .828 | .842 | **.849** | .845 | .825 | .835 |
| MiddlePhalanxTW | .597 | .610 | .558 | .610 | **.630** | .623 | .571 | .623 |
| MoteStrain | .883 | **.884** | .871 | .871 | .879 | .883 | .872 | .871 |
| OSULeaf | .603 | .616 | .645 | .632 | **.661** | .653 | .645 | .636 |
| PenDigits | .977 | .981 | .985 | .984 | **.985** | .985 | .983 | .984 |
| PhalangesOutlinesCorrect | **.850** | .848 | .837 | .839 | .840 | .847 | .832 | .837 |
| PickupGestureWiimoteZ | **.760** | .700 | .740 | **.760** | .740 | .720 | .740 | .740 |
| Plane | **1.000** | .990 | **1.000** | .990 | .990 | .990 | **1.000** | .990 |
| PowerCons | .989 | **1.000** | .989 | .989 | **1.000** | **1.000** | .989 | **1.000** |
| ProximalPhalanxOutlineAgeGroup | .859 | .859 | .859 | .854 | .844 | .844 | **.863** | .854 |
| ProximalPhalanxOutlineCorrect | .897 | **.907** | .890 | **.907** | .904 | **.907** | .887 | .904 |
| ProximalPhalanxTW | .790 | .815 | .815 | .790 | **.820** | .820 | **.820** | .785 |
| RacketSports | .842 | .875 | **.895** | .888 | .875 | .862 | **.895** | .888 |
| ShakeGestureWiimoteZ | .760 | .740 | .800 | **.900** | **.900** | .880 | .760 | **.900** |
| ShapeletSim | **.489** | .483 | .483 | .478 | **.489** | **.489** | .478 | .478 |
| SmoothSubspace | **1.000** | **1.000** | **1.000** | **1.000** | **1.000** | **1.000** | **1.000** | **1.000** |
| SonyAIBORobotSurface1 | .819 | **.822** | .750 | .772 | .809 | .812 | .747 | .764 |
| SonyAIBORobotSurface2 | .821 | **.834** | .824 | .827 | .832 | .831 | **.834** | .833 |
| Strawberry | **.981** | **.981** | .976 | .978 | **.981** | **.981** | .978 | .976 |
| Symbols | .896 | .890 | .865 | .893 | **.899** | **.899** | .862 | .885 |
| SyntheticControl | .990 | .987 | .993 | .997 | .993 | **1.000** | .993 | .997 |
| ToeSegmentation1 | .575 | .575 | .649 | .605 | **.662** | .636 | .645 | .601 |
| ToeSegmentation2 | .669 | .700 | .892 | .900 | **.908** | **.908** | .900 | .892 |
| Trace | .950 | .960 | **.990** | **.990** | **.990** | **.990** | **.990** | **.990** |
| TwoLeadECG | .904 | .944 | .903 | .895 | **.946** | .944 | .910 | .903 |
| TwoPatterns | .981 | **.998** | .984 | .997 | .987 | **.998** | .984 | .997 |
| UMD | **1.000** | **1.000** | **1.000** | **1.000** | **1.000** | **1.000** | **1.000** | **1.000** |
| UWaveGestureLibraryX | .804 | .812 | .811 | **.814** | .811 | .812 | .810 | .814 |
| UWaveGestureLibraryY | .707 | .724 | .704 | **.724** | .707 | .723 | .702 | **.724** |
| UWaveGestureLibraryZ | .744 | **.763** | .727 | .757 | .751 | .760 | .727 | .757 |
| Wafer | .995 | .995 | .997 | .997 | .998 | .997 | .997 | **.998** |
| Wine | **.796** | .778 | .611 | .648 | .704 | .704 | .630 | .648 |
| Yoga | **.875** | .865 | .864 | .863 | .860 | .862 | .863 | .866 |
| *Mean* | .844 | .852 | .838 | .849 | .856 | **.858** | .837 | .849 |

## F. Datasets excluded from the UCR/UEA archives

*Table 5.* Datasets from the UCR and UEA archives excluded from our evaluation because they fall outside TabPFN's operating regime ($K{>}10$ and/or $m{\cdot}T{>}500$). Totals: 83 excluded; 75 retained.

| Dataset | Archive | K | m | T | m·T | Exclusion reason |
|---|---|---|---|---|---|---|
| ACSF1 | UCR | 10 | 1 | 1460 | 1460 | $m{\cdot}T{=}1460$ |
| Adiac | UCR | 37 | 1 | 176 | 176 | $K{=}37$ |
| BeetleFly | UCR | 2 | 1 | 512 | 512 | $m{\cdot}T{=}512$ |
| BirdChicken | UCR | 2 | 1 | 512 | 512 | $m{\cdot}T{=}512$ |
| Car | UCR | 4 | 1 | 577 | 577 | $m{\cdot}T{=}577$ |
| CinCECGTorso | UCR | 4 | 1 | 1639 | 1639 | $m{\cdot}T{=}1639$ |
| Computers | UCR | 2 | 1 | 720 | 720 | $m{\cdot}T{=}720$ |
| CricketX | UCR | 12 | 1 | 300 | 300 | $K{=}12$ |
| CricketY | UCR | 12 | 1 | 300 | 300 | $K{=}12$ |
| CricketZ | UCR | 12 | 1 | 300 | 300 | $K{=}12$ |
| Crop | UCR | 24 | 1 | 46 | 46 | $K{=}24$ |
| EOGHorizontalSignal | UCR | 12 | 1 | 1250 | 1250 | $K{=}12, m{\cdot}T{=}1250$ |
| EOGVerticalSignal | UCR | 12 | 1 | 1250 | 1250 | $K{=}12, m{\cdot}T{=}1250$ |
| Earthquakes | UCR | 2 | 1 | 512 | 512 | $m{\cdot}T{=}512$ |
| EthanolLevel | UCR | 4 | 1 | 1751 | 1751 | $m{\cdot}T{=}1751$ |
| FaceAll | UCR | 14 | 1 | 131 | 131 | $K{=}14$ |
| FacesUCR | UCR | 14 | 1 | 131 | 131 | $K{=}14$ |
| FiftyWords | UCR | 50 | 1 | 270 | 270 | $K{=}50$ |
| Fungi | UCR | 18 | 1 | 201 | 201 | $K{=}18$ |
| GestureMidAirD1 | UCR | 26 | 1 | 360 | 360 | $K{=}26$ |
| GestureMidAirD2 | UCR | 26 | 1 | 360 | 360 | $K{=}26$ |
| GestureMidAirD3 | UCR | 26 | 1 | 360 | 360 | $K{=}26$ |

*(continued)*

| Dataset | Archive | K | m | T | m·T | Exclusion reason |
|---|---|---|---|---|---|---|
| HandOutlines | UCR | 2 | 1 | 2709 | 2709 | $m{\cdot}T{=}2709$ |
| Haptics | UCR | 5 | 1 | 1092 | 1092 | $m{\cdot}T{=}1092$ |
| Herring | UCR | 2 | 1 | 512 | 512 | $m{\cdot}T{=}512$ |
| HouseTwenty | UCR | 2 | 1 | 2000 | 2000 | $m{\cdot}T{=}2000$ |
| InlineSkate | UCR | 7 | 1 | 1882 | 1882 | $m{\cdot}T{=}1882$ |
| InsectEPGRegularTrain | UCR | 3 | 1 | 601 | 601 | $m{\cdot}T{=}601$ |
| InsectEPGSmallTrain | UCR | 3 | 1 | 601 | 601 | $m{\cdot}T{=}601$ |
| InsectWingbeatSound | UCR | 11 | 1 | 256 | 256 | $K{=}11$ |
| LargeKitchenAppliances | UCR | 3 | 1 | 720 | 720 | $m{\cdot}T{=}720$ |
| Lightning2 | UCR | 2 | 1 | 637 | 637 | $m{\cdot}T{=}637$ |
| Mallat | UCR | 8 | 1 | 1024 | 1024 | $m{\cdot}T{=}1024$ |
| MixedShapesRegularTrain | UCR | 5 | 1 | 1024 | 1024 | $m{\cdot}T{=}1024$ |
| MixedShapesSmallTrain | UCR | 5 | 1 | 1024 | 1024 | $m{\cdot}T{=}1024$ |
| NonInvasiveFetalECGThorax1 | UCR | 42 | 1 | 750 | 750 | $K{=}42, m{\cdot}T{=}750$ |
| NonInvasiveFetalECGThorax2 | UCR | 42 | 1 | 750 | 750 | $K{=}42, m{\cdot}T{=}750$ |
| OliveOil | UCR | 4 | 1 | 570 | 570 | $m{\cdot}T{=}570$ |
| PLAID | UCR | 11 | 1 | 1345 | 1345 | $K{=}11, m{\cdot}T{=}1345$ |
| Phoneme | UCR | 39 | 1 | 1024 | 1024 | $K{=}39, m{\cdot}T{=}1024$ |
| PigAirwayPressure | UCR | 52 | 1 | 2000 | 2000 | $K{=}52, m{\cdot}T{=}2000$ |
| PigArtPressure | UCR | 52 | 1 | 2000 | 2000 | $K{=}52, m{\cdot}T{=}2000$ |
| PigCVP | UCR | 52 | 1 | 2000 | 2000 | $K{=}52, m{\cdot}T{=}2000$ |
| RefrigerationDevices | UCR | 3 | 1 | 720 | 720 | $m{\cdot}T{=}720$ |
| Rock | UCR | 4 | 1 | 2844 | 2844 | $m{\cdot}T{=}2844$ |
| ScreenType | UCR | 3 | 1 | 720 | 720 | $m{\cdot}T{=}720$ |
| SemgHandGenderCh2 | UCR | 2 | 1 | 1500 | 1500 | $m{\cdot}T{=}1500$ |
| SemgHandMovementCh2 | UCR | 6 | 1 | 1500 | 1500 | $m{\cdot}T{=}1500$ |
| SemgHandSubjectCh2 | UCR | 5 | 1 | 1500 | 1500 | $m{\cdot}T{=}1500$ |
| ShapesAll | UCR | 60 | 1 | 512 | 512 | $K{=}60, m{\cdot}T{=}512$ |
| SmallKitchenAppliances | UCR | 3 | 1 | 720 | 720 | $m{\cdot}T{=}720$ |
| StarLightCurves | UCR | 3 | 1 | 1024 | 1024 | $m{\cdot}T{=}1024$ |
| SwedishLeaf | UCR | 15 | 1 | 128 | 128 | $K{=}15$ |
| UWaveGestureLibraryAll | UCR | 8 | 1 | 945 | 945 | $m{\cdot}T{=}945$ |
| WordSynonyms | UCR | 25 | 1 | 270 | 270 | $K{=}25$ |
| Worms | UCR | 5 | 1 | 900 | 900 | $m{\cdot}T{=}900$ |
| WormsTwoClass | UCR | 2 | 1 | 900 | 900 | $m{\cdot}T{=}900$ |
| ArticularyWordRecognition | UEA | 25 | 9 | 144 | 1296 | $K{=}25, m{\cdot}T{=}1296$ |
| AtrialFibrillation | UEA | 3 | 2 | 640 | 1280 | $m{\cdot}T{=}1280$ |
| BasicMotions | UEA | 4 | 6 | 100 | 600 | $m{\cdot}T{=}600$ |
| CharacterTrajectories | UEA | 20 | 3 | 119 | 357 | $K{=}20$ |
| Cricket | UEA | 12 | 6 | 1197 | 7182 | $K{=}12, m{\cdot}T{=}7182$ |
| DuckDuckGeese | UEA | 5 | 1345 | 270 | 363150 | $m{\cdot}T{=}363150$ |
| EigenWorms | UEA | 5 | 6 | 17984 | 107904 | $m{\cdot}T{=}107904$ |
| Epilepsy | UEA | 4 | 3 | 206 | 618 | $m{\cdot}T{=}618$ |
| EthanolConcentration | UEA | 4 | 3 | 1751 | 5253 | $m{\cdot}T{=}5253$ |
| FaceDetection | UEA | 2 | 144 | 62 | 8928 | $m{\cdot}T{=}8928$ |
| FingerMovements | UEA | 2 | 28 | 50 | 1400 | $m{\cdot}T{=}1400$ |
| HandMovementDirection | UEA | 4 | 10 | 400 | 4000 | $m{\cdot}T{=}4000$ |
| Handwriting | UEA | 26 | 3 | 152 | 456 | $K{=}26$ |
| Heartbeat | UEA | 2 | 61 | 405 | 24705 | $m{\cdot}T{=}24705$ |
| InsectWingbeat | UEA | 10 | 200 | 20 | 4000 | $m{\cdot}T{=}4000$ |
| LSST | UEA | 14 | 6 | 36 | 216 | $K{=}14$ |
| Libras | UEA | 15 | 2 | 45 | 90 | $K{=}15$ |
| MotorImagery | UEA | 2 | 64 | 3000 | 192000 | $m{\cdot}T{=}192000$ |
| NATOPS | UEA | 6 | 24 | 51 | 1224 | $m{\cdot}T{=}1224$ |
| PEMS-SF | UEA | 7 | 963 | 144 | 138672 | $m{\cdot}T{=}138672$ |
| PhonemeSpectra | UEA | 39 | 11 | 217 | 2387 | $K{=}39, m{\cdot}T{=}2387$ |
| SelfRegulationSCP1 | UEA | 2 | 6 | 896 | 5376 | $m{\cdot}T{=}5376$ |
| SelfRegulationSCP2 | UEA | 2 | 7 | 1152 | 8064 | $m{\cdot}T{=}8064$ |
| SpokenArabicDigits | UEA | 10 | 13 | 65 | 845 | $m{\cdot}T{=}845$ |
| StandWalkJump | UEA | 3 | 4 | 2500 | 10000 | $m{\cdot}T{=}10000$ |
| UWaveGestureLibrary | UEA | 8 | 3 | 315 | 945 | $m{\cdot}T{=}945$ |

