# OpenReview forum: "A Causal DAG Prior for Synthetic Time-Series Classification Datasets"
_ICML.cc/2026/Workshop/FMSD — FMSD @ ICML 2026 Poster_

### Official Review · Reviewer_AtKz · 2026-05-12
**Review of “A Causal DAG Prior for Synthetic Time-Series Classification Datasets”**

**Rating:** 7
**Confidence:** 5

**Review:**

## Summary

This paper proposes a causal DAG prior for generating synthetic labelled time-series classification datasets. The key idea is to build a unified graph whose nodes can be either tabular variables or time-series trajectories, allowing causal relations across modalities, channels, timesteps, and labels. The authors use the generated datasets to fine-tune TabPFN v2.5 and show improvements over the original model and a tabular-only ablation.

## Strengths

The main strength of the paper is its creative use of a single causal DAG to model both tabular and time-series modalities. This is a novel and interesting direction for synthetic data generation.
The proposed prior provides a valuable attempt to bridge tabular foundation models and time-series classification, especially by explicitly modeling cross-modal causal structure.

The experimental comparison with a tabular-only ablation is useful, as it helps isolate the contribution of the time series specific components of the generator.

## Areas for Improvement

The motivation for the label generation mechanism is not sufficiently clear. Since labels are obtained from discretized tabular nodes, the paper should better justify why this produces meaningful and realistic classification targets for time series tasks.

The validity of the generated labels also needs stronger empirical evidence. Additional analyses on class separability, temporal relevance, or the dependence between generated time series and labels would make the argument more convincing.

Fine tuning TabPFN is a useful first validation, but it is not sufficient to fully demonstrate the value of the proposed prior. A stronger pretraining level evaluation, following settings similar to TiCT or TIC-FM [1], would better show whether the prior can support training a time series classification foundation model.

The writing would benefit from further polishing. Some design choices, motivations, and experimental interpretations should be explained more clearly.


[1]: Fang, Jun et al. “Rethinking Zero-Shot Time Series Classification: From Task-specific Classifiers to In-Context Inference.” ArXiv abs/2602.00620 (2026): n. pag.

---

### Official Review · Reviewer_MeSJ · 2026-05-21
**A Promising Synthetic Causal DAG Prior for Time-Series Classification**

**Rating:** 8
**Confidence:** 4

**Review:**

## **Summary of contributions**

The paper proposes a Causal DAG Prior for Synthetic Time-Series Classification Datasets.
The prior is constructed based on randomly sampled DAGs over tabular and series nodes. The proposed framework generates complete datasets (X, y) with cross-modal causal structure across channels, timesteps, and labels, addressing a setting not covered by existing synthetic priors. Labels are obtained through discretization of tabular nodes. The method is evaluated using and finetuning TabPFN v2.5 with minimal architectural modifications in order to isolate the impact of the proposed prior. Furthermore, the paper introduces an input preprocessing step that converts the signals into short temporal patches (similarly to strategies commonly used in time-series forecasting) and applies channel-wise normalization. Experiments are conducted on subsets of the UCR and UEA benchmarks.
The evaluation compares raw TabPFNv2.5, preprocessing only, finetuning with preprocessing, and variants without temporal series nodes. Results indicate that combining preprocessing with the proposed causal prior yields the best performance, while removing series nodes in the prior significantly degrades results, highlighting the importance of temporal structure in the synthetic prior.

## **Strengths and weaknesses**
### **Strengths**
- Clear focus on an underexplored issue: synthetic data design for TSC under ICL.
- Interesting and original causal prior combining tabular and time-series modalities within a unified DAG framework.
- The paper is well written and addresses a topic of clear interest to the community. The proposed method is clearly explained, and the experimental setup is clear.
### **Weaknesses**
- The paper does not compare against methods specifically designed for time series classification. It would be valuable to include a few comparisons with classical supervised TSC baselines trained directly on the target datasets, and report metrics (accuracy and ROC-AUC) to better contextualize the performance of the proposed approach.
## **Suggestions**
- Include comparisons with standard supervised TSC baselines to address the limitation mentioned above.

---

### Official Review · Reviewer_EkG5 · 2026-05-21
**A Causal DAG Prior for Synthetic Time-Series Classification Datasets**

**Rating:** 7
**Confidence:** 3

**Review:**

## Summary

The paper introduces a causal-DAG synthetic data generator for pretraining time-series classifiers, sampling typed nodes (tabular scalars and series-valued trajectories) with edges spanning small NNs, Conv1D, and discretisation to produce labelled multivariate TSC datasets with cross-modal causal structure. They probe the generator by finetuning TabPFN v2.5 with minimal input/output edits and show it significantly outperforms both the upstream model and a tabular-only ablation on 75 UCR/UEA datasets.

## Strengths

- Clean and well-motivated contribution: a generator that produces complete labelled datasets with cross-modal tabular and series causal structure fills a real gap.
- Honest scoping : the limitations section openly acknowledges this is a generator paper, not a SOTA TSC foundation model, which helps calibrate reviewer expectations.


## Areas for Improvement

- No comparison to existing synthetic time-series generators (KernelSynth, CauKer, TiCT) under matched conditions, so it is unclear whether the gain comes from cross-modal causal structure specifically or just from any reasonable time-series prior beyond the tabular one.
- No comparison to standard TSC baselines (ROCKET, HIVE-COTE, InceptionTime), so even within the 75 datasets the reader cannot calibrate whether finetuned TabPFN with this generator is competitive with the broader TSC literature.